# Implementing Supported Self-Management in Community-Based Stroke Care: A Secondary Analysis of Nurses’ Perspectives

**DOI:** 10.3390/jcm9040985

**Published:** 2020-04-01

**Authors:** Lisa Kidd, Joanne Booth, Maggie Lawrence, Anne Rowat

**Affiliations:** 1Nursing & Healthcare, School of Medicine, Dentistry & Nursing, University of Glasgow; Glasgow G12 8LL, UK; 2School of Health & Life Sciences, Glasgow Caledonian University; Glasgow G4 0BA, UK; jo.booth@gcu.ac.uk (J.B.); Maggie.lawrence@gcu.ac.uk (M.L.); 3School of Health & Social Care, Edinburgh Napier University; Edinburgh EH11 4DY, UK; a.rowat@napier.ac.uk

**Keywords:** supported self-management, implementation, stroke, nurses, person-centred, qualitative research

## Abstract

The provision of supported self-management (SSM) is recommended in contemporary guidelines to address the longer-term needs and outcomes of stroke survivors and their families, yet its implementation across stroke pathways has been inconsistent. This paper presents a secondary analysis of qualitative data, which aims to identify and offer insight into the challenges of implementing SSM from the perspectives of community stroke nurses (*n* = 14). The findings revealed that the implementation of SSM in stroke is influenced by factors operating at multiple levels of the healthcare system. Contextual challenges arise because of different understandings and interpretations of what SSM is, what it comprises and professionals’ perceptions of their roles in its implementation in practice. A professionally controlled, one-size-fits-all model of SSM continues to be reinforced within organizations, offering few opportunities for nurses to deliver contextually tailored and person-centred SSM. In conclusion, there are many professional concerns and organizational tensions that need to be addressed across multiple layers of the healthcare system to achieve the consistent implementation of contextually tailored and person-centred SSM following a stroke. Attempts to address these challenges will help to narrow the gap between policy and practice of implementing SSM, ensuring that stroke survivors and families benefit from SSM in the longer-term.

## 1. Introduction

Stroke is recognized as an acute event followed by long-term treatments and support that facilitate rehabilitation and prevent further stroke. In the United Kingdom (UK), strokes cost the economy £26 billion each year; much of which is the direct cost to stroke survivors and their families [1]. There are currently up to 1.2 million stroke survivors living in the UK [1]. This figure is predicted to rise to 1.5 million by the year 2025 [1]. Many stroke survivors experience significant and complex physical, cognitive and emotional stroke-related impairments, and multimorbidities that require longer-term management, and meaningful and effective support in primary and community care settings [1,2,3]. The identification and management of the long-term consequences of stroke is not only a priority issue in the UK but also a global priority for stroke researchers [4,5].

Supported self-management (SSM) in stroke is increasingly recognized as the preferred term for a multidisciplinary team approach, which involves working in partnership with stroke survivors and families in an ongoing and sustained manner. The focus is on supporting people to develop the capacity and confidence to manage the impact of their stroke in the context of their lives in a way that is meaningful and purposeful for them, and to ‘be in charge’ of their own self-management decision making [6,7,8,9]. Contemporary notions of SSM position it as a move away from the traditional perceptions of education—compliance with professionally driven treatment plans and goals, and a focus on function and level of impairment—because this does not ‘fit’ with how people with long-term conditions typically perceive their own engagement in self-management [6,10,11].

The integration of SSM into post-discharge rehabilitation pathways for stroke survivors has been recommended in European and UK-wide policy documents and clinical guidelines for addressing the longer-term needs and outcomes of stroke survivors, and is widely welcomed and valued by stroke survivors and their families [12,13,14,15,16,17,18,19,20]. Evidence from international clinical trials shows that SSM improves stroke survivors’ quality of life and self-efficacy, promotes engagement in healthy lifestyle behaviors, and potentially reduces the use of health services [6,7,8,9,21,22]. SSM should feature as a core component in post-stroke rehabilitation [23]. Stroke rehabilitation is a time-limited resource, where individuals often feel abandoned once therapy finishes [23]. The integration of SSM during rehabilitation could provide a shared platform for collaboration from the start of therapy, facilitating transitions to community and longer-term support [23]. However, SSM has different identities [24]. SSM is sometimes viewed as a strategy for optimizing secondary stroke prevention, where the focus is on stroke survivors’ adherence to medical management and reducing risk factors for stroke, and as an approach for increasing stroke survivors’ activity levels and adherence to rehabilitative self-exercise [24]. Others view SSM as a broader approach for enhancing stroke survivors’ participation in society, reengagement with desired or valued roles, and focused on bolstering individuals’ resilience and resourcefulness in rebuilding their lives after stroke in the long-term [24]. These different interpretations, and the differences in the level of control, ownership or independence the individual has in their relationships with rehabilitation professionals, cause confusion over what SSM is and where it ‘fits’ within existing systems of stroke rehabilitation. Thus, unsurprisingly, recent UK audit data has confirmed that SSM implementation within community-based rehabilitation in the UK is variable and sub-optimal [25]. As a result, stroke survivors continue to describe a lack of support for person-centred self-management in the longer-term within current service provision [15,26].

Despite the range of ‘self-management interventions’ in stroke [6,7,8,9,21,22], these do not neatly transfer, become readily implemented, or have sustained effects across different geographical and organizational contexts of community rehabilitation practice, and there is no established or ‘gold standard’ model of stroke SSM [11,19]. The success of complex interventions in real-world settings is influenced by a wide range of contextual factors. These include the external and internal infrastructure and the organization in which the intervention is being implemented, as well as the skills, perceptions and values of the people involved in its delivery, and the ‘fit’ of the intervention with practitioners’ current practice and workloads [27]. Understanding and addressing these contextual factors is therefore crucial to the effective implementation of complex interventions such as SSM for people with stroke, yet the influence of context is still poorly understood [28,29]. The objective of this paper is to describe nurses’ perspectives and experiences of implementing SSM within the context of community-based stroke rehabilitation provision. The aim is to identify potential challenges, operating at multiple levels of the healthcare system, which could inform future research and practice on implementing SSM in a timely, appropriate and sustainable way.

## 2. Materials and Methods

### 2.1. Study Design and Data Source

This paper reports on the findings of a secondary analysis of qualitative data. Secondary analysis of qualitative research data offers an opportunity to bring a new perspective to, or ask new research questions of, existing data, collected for a different primary purpose, to inform and guide future research [30]. The data for this secondary analysis were originally drawn from an evaluation study which aimed to design, develop and evaluate the feasibility and acceptability of a nurse-facilitated, tailored stroke SSM intervention in the UK and is reported in full elsewhere [11]. Briefly, the intervention, which was delivered to stroke survivors who were up to 12 months post-stroke, comprised a collaborative, personalized self-management conversation with stroke nurses, underpinned by motivational interviewing and the use of the Patient Activation Measure [31], to develop a theoretically tailored self-management action plan.

### 2.2. Study Participants

Participants (*n* = 14) across the two-phase study included community-based stroke nurses working in three NHS Boards in Scotland (called Trusts in England). The three NHS Boards were selected as they were broadly representative of Scotland’s 15 NHS Boards, offering a mix of demographic, geographical and organizational contexts. No exclusion criteria in terms of grade, previous experience or length of time qualified were applied to the sample. Willing nurses were invited to participate, and informed consent obtained. The demographics of the study participants are shown in Table 1.

### 2.3. Study Instruments (Focus Groups and Telephone Interviews)

Data were collected in 2013 via focus groups (*n* = 2) and individual interviews (*n* = 2, where distance permitted a focus group being possible within the original study timeframe). Focus groups and interviews were conducted by LK (and the project research assistant) and lasted 1–1.5 h. The focus groups were conducted near nurses’ place of work and the interviews were conducted by telephone. The focus groups and interviews followed an interview guide which asked nurses about their experiences of delivering the SSM in practice and their views on issues related to its design, delivery and future refinement, and its implementation and embedding within the scope of their clinical practice e.g., What does self-management mean to you? How easy did you find it to use the intervention in practice? How did using the intervention impact upon your attitudes or thoughts towards self-management? How did using the intervention impact on your care delivery practices in providing support for self-management? What changes need to be made to the intervention to make it work most effectively in your practice?

### 2.4. Data Analysis

All interviews were audio-recorded, transcribed verbatim and thematically analyzed by hand to identify key issues and themes within the data. To ensure credibility, relevance, transparency and trustworthiness of the findings, the emerging codes and themes were consistently questioned and reviewed by members of the research team (L.K., J.B., M.L., A.R.) and a stroke survivor research advisory group. These discussions, which prompted an iterative process of working between the coding frame and original transcripts, helped the research team to feel confident that thematic saturation had been achieved in addressing the original study aim. The original study was completed in 2014 and ethical approval was received from the West of Scotland Research Ethics Committee and NHS Research Scotland Permissions Coordinating Centre, respectively.

### 2.5. Secondary Analysis

The secondary analysis reported on here aimed to explore the following research question: What contextual factors influence how SSM is interpreted and implemented by stroke nurses in practice? This question was broader than in the original study, which focused on the design and evaluation of one specific SSM intervention. We believed that the ‘fit’ between the currency and depth of the existing dataset and the ‘new’ research question, however, was appropriate to gain a broader insight into the challenges associated with implementing SSM in practice. The dataset included transcripts from three focus groups and two telephone interviews. Each transcript was assessed for the quality of the data, ensuring it provided coherent information that aligned with the new research question (L.K., J.B., M.L., A.R.). All transcripts were thematically re-analysed using Braun and Clarke’s framework [32] (L.K., J.B., M.L., A.R.). Lau et al.’s [27] framework for understanding the contextual barriers and facilitators to the implementation of interventions in primary care was used to underpin the analysis (Figure 1).

## 3. Results

The results are framed by, and presented to align with, the components of Lau et al.’s [27] framework. The alignment of the themes and specific example quotations are described in the following narrative. Verbatim quotations have been included to exemplify the themes where participants are referred to using a unique number (e.g., N1, N2, N3; N denoting ‘nurse’). To protect confidentiality and anonymity, the health board pertaining to where nurses worked has not been included in the unique identifier.

### 3.1. External Context

The external context referred to the presence or absence of legislative or local policies, outcome frameworks, strategy documents, the presence of dominant paradigms, stakeholder buy in, and the external infrastructure [27]. A few of the nurses spoke about the presence or absence of legislative and local policies, guidelines and local or national initiatives designed to support the implementation of SSM or nurses’ roles within SSM at the time of the study. Despite the intensifying political support for SSM at the time of the study, there was a sense amongst many of the nurses that SSM wasn’t anything ‘new’ to them and was simply putting a different ‘label’ on the same thing. Nurses did not frequently distinguish SSM from other aspects of their nursing care.

“I don’t know enough about it (what the policy says) but I have been discussing it (SSM) with patients and referring patients and whatever and I’m sitting here thinking this is what we do anyway…I think we are kind of doing it you know?”(N2)

“It is my job…a chief part of my job is promoting self-management…and hoping that the end result would be that the patient would be able to self-manage at the end of our advice and support period.”(N5)

Some of the nurses commented that guidance on implementing SSM in practice would only be useful for inexperienced nurses or someone new in post.

“I suppose for an inexperienced nurse (guidance on implementing SMS) would be useful but I think when you have been doing the job for as long as I have then it’s kind of natural you know, you wouldn’t really need a tool because you are able to know yourself with the experience that you have got. I wouldn’t feel that I needed (to use an assessment tool) because I just naturally ask things as anyway as part of my assessment.”(N6)

For others, SSM involved working in a different way to how they did previously and the acquisition and development of new skills. However, they noted that this approach, which involved genuine partnership working with stroke survivors, arguably took longer to ‘do’ and was at odds with the design and complexity of the healthcare system, where the dominant paradigm inherent in the external context (which filtered down to the organization) appeared to be one of efficiency rather than person-centred care.

“There is obviously pressure to get people through the system in a quicker way and being able to self-manage. That arguably takes time to get folk to the stage so that doesn’t quite fit with the model of ‘getting in and getting out’ in a very short time and I think communication and therapeutic relationships are really important.”(N4)

Consequently, nurses perceived that the external infrastructure was not always conducive to supporting the implementation of SSM in a timely and consistent way. Several of the nurses expressed frustration at the lack of joined-up working between, or challenging bureaucratic processes involved in accessing, external health- and social-care-related organizations, and access to and sustainability of community groups, resources and ‘assets’ within local communities. This was particularly difficult in remote and rural areas, where the centralization of services meant that outlying communities did not benefit as much.

“What hinders us is the lack of resources and services that are in (NHS Board name) and how remote our caseload is. You’ve got some people that live in (city) that have got access to services everywhere which is fantastic…then you’ve got patients fifty miles away that can’t access anything. Small villages and towns which don’t offer anything at all is a severe barrier (to implementing SMS).”(N6)

The political ‘flux’ triggered by a frequently changing political landscape where existing policies and plans for service delivery could be rapidly abolished or replaced by something different, led to nurses feeling a sense of demotivation or mistrust, and a reluctance to ‘buy in’ to SSM if it involved having to change their current practices.

“It is confusing what’s out there though because every political boundary you cross over be it north, south, there will always be this bickering between them and you get better this and that but there’s so much out there that you really don’t know everything that is available and then maybe if a different government comes in they will change the boundaries again and they will change their names. You think you have this knowledge up here (pointing to head) and then you go back to get the telephone number and the contact that you had, and they have all changed.”(N7)

### 3.2. Organisation

Organization referred to aspects such as the presence of a positive culture of support, clinical leadership, and organizational readiness to support implementation efforts, as well as available resources and processes and systems, and good relationships to facilitate this implementation [27]. A positive culture of support for SSM appeared to be characterized by having good clinical leadership or champions for self-management within the organization as well as good teamworking, where staff could share experiences and learn from and support each other. This helped to promote a shared vision of SSM and reduce professional isolation.

“That’s one thing with this team, we are all quite happy. We’ve been doing the same job for a long time. I think there’s a lot of clinical support here so you can bounce ideas off people because it’s difficult at times. We phone each other up; any queries we would just ask each other.”(N10)

This level of organizational and collegiate support, however, was not consistent across NHS Boards, potentially contributing to siloed working.

“It’s not consistent though…if you spoke to some other areas some stroke nurses do feel very isolated with a caseload and maybe even as part of a team. They’re having to cope with sometimes very challenging issues.”(N12)

In relation to organizational readiness, the processes and systems within the provision of community stroke care did not appear to align with, or support nurses in, the delivery of ‘person-led SSM. For example, the delivery of care and support following acute discharge and the end of formal rehabilitation therapy (for most individuals) was structured around a predetermined number of visits (usually over the course of a year). The structure and frequency of their visits varied little in accordance with individuals’ specific rehabilitation therapy regimes and their self-management needs and experiences. It is unclear from the nurses’ accounts how the nurses distinguished SSM as being different from other aspects of nursing care and support that they were providing. However, several of the nurses interpreted SSM as synonymous with ‘going it alone and ‘preparation for discharge’, when stroke survivors would be discharged from further formal nursing follow up. As a result, SSM did not appear to be considered as an integrated and embedded component of their support from the very start of their interactions with stroke survivors in the community.

“Our follow up is a year post discharge so I tell them what we would like to achieve by the end of that year and then I maybe see them five times in that year. It’s (SMS) relevant all the way through (SMS) but I don’t do much at the start. I would be seeing them again three months post discharge and I would say it is quite relevant then. It’s very difficult for when we come to discharge a patient, you will find that they will keep contacting you you know, and that’s not really what we are about. Basically, we are about self-management which means that they would need to be able to go and contact the resources and support services themselves. The end result of (my job) would be that the patient would be able to self-manage by the end of our advice and support period.”(N5)

The lack of additional resources and capacity within the organization to support and enable nurses to reflect on their current practice and to work collaboratively with stroke survivors and their families led to some concerns that to work in this way would create more pressure on an already time- and resource-stretched system. SSM was viewed by some nurses as an additional task that was going to have implications for their time management and workload.

“All the information that we’ve (given) about SSM, it’s really time consuming and everything that we tend to do…we’re on a timescale all day…and you’ve got so many visits, you need to try and get back for a clinic and I’m just constantly watching the time. We can easily be there half an hour, three quarters of an hour sometimes…but you can be there longer…and you come out and you still haven’t done everything that you need to do. I’m just thinking of the time that it takes to realistically do this with (SSM) with so many patients.”(N2)

“What we do ourselves is a lot you know so by the time we have gone over everything that we would like to offer after they have just had a stroke and then having to start talking to them about (self-management), then it’s going to affect our workload and the time that we have. It will impact on the length of visits in that we can’t fit as many people in. If it’s something we are going to have to facilitate, it’s going to have an impact on our time.”(N5)

### 3.3. Professionals

Professionals related to nurses’ perceptions of their professional roles, identities and professionalism, concerns about autonomy and confidence practicing SSM, and attitudes towards implementing SSM [27]. Although SSM was viewed as important, working more collaboratively with stroke survivors and families was seen by nurses as a potential threat and challenged their perceived roles which were to ‘do for’ and ‘to fix’ or their professional knowledge and experience.

“We have guidelines that we want them to be able to achieve, given our experience and knowledge of stroke patients. We have an idea of what we want that patient to be able to do. We have to decide if they are being realistic you know or unrealistic, I suppose. Sometimes people think that they need things they don’t. Sometimes people are very unrealistic.”(N5)

“As nurses, we’ve not been used to patients self-managing, so we’ve probably put a lot of people in that role (of being dependent) because we have to make it better. We were programmed to step in there and do everything from the basics to washing, right up to dressings and things and to stand back, it’s difficult. They have to be able to do that (be independent) and we (nurses) have to let them go and let them be independent themselves”(N10)

“I think that stroke nurses perhaps have to recognize that you can’t fix everything. I would say when we started out in this role, we went out to fix it. It’s gradually changing but that’s the biggest problem and always was; you want to fix it.”(N12)

Stepping outside of these ‘safe’ and ‘familiar’ roles was difficult and uncertain, which could reduce nurses’ confidence in challenging the ‘status quo’ in their practice. The nurses frequently alluded to ‘professional gatekeeping’, where they decided who was suitable for ‘self-management, with people labelled as ‘self-managers’ or ‘non self-managers’.

“We have to…decide if they (people) are being realistic you know, or are they being unrealistic I suppose. Sometimes people think that they need things they don’t.”(N5)

“You’ve got people that are motivated and want to do it (SSM) and you’ve got other people that don’t and that’s what they’ve been like prior to their stroke so they are not going to move on”(N10)

There appeared to be a lack of confidence amongst the nurses in ‘letting’ stroke survivors engage in self-management on their own, as this was perceived as ‘risky’ and causing more harm than good. Such perceptions and fears may stem in part from working in an organizational model that prioritizes patient safety and risk reduction over patient (and nurse) autonomy. As a result, goal setting and self-management decision making remained largely within the control of professionals rather than being person-led.

“People will try and test them out (self-management strategies). As long as it’s safe, sometimes you have to allow them to try that and you know, actually they will learn from it”(N4)

“We have had this one man that wanted to get back and ride a horse or a motorbike and his expectations were that he would be able to do that and give him his due, at the end of it he had an adapted quad bike and things. He raised the level far higher than any therapist would have through that he would have achieved because we err on the side of risk assessment.”(N10)

### 3.4. Intervention

Intervention referred to the nature and characteristics of the intervention, its complexity and practicality, customization of the intervention, and its impact [27]. Many of the ideas within this thematic category overlapped with those already discussed, e.g., the fit of the ethos of SSM with the organization (relating to customization, for example). Within this theme, however, nurses referred to varying interpretations of SSM (more broadly, rather than just as one intervention). The nature and scope of what a package of SSM comprised or ‘looked like’ was also spoken about. From this, it was clear that SSM was a broad, nebulous concept, with several nurses unsure as to what it really involved, which may have contributed to a lack of ‘buy in’ for SSM. For some, SSM was about ‘living well’ and characterized by shared responsibility whilst, for others, it was viewed as ‘compliance’ or ‘education’.

“A lot of it is educating…educating the patient on what we would say are the risk factors so that they have knowledge of why they should be helping themselves to stop smoking, drink less and take more activity.”(N8)

“It (SSM) is about shared responsibility. I think sometimes what self-management is not is when people think it’s about getting on with it yourself and on you go. It’s very much about support and shared support. It’s about personal responsibility and being part of that and to a certain extent in control of the choices and options you make on your recovery.”(N11)

In response to the ambiguity about SSM, many nurses favored more familiar and traditional types of post-stroke support, such as secondary prevention and lifestyle behavior change messages.

“If it’s the likes of smoking cessation, they will follow through if they are ready to do that, but I suppose that is a nice and neat intervention on a specific issue.”(N4)

Nurses’ language around SSM reinforced the idea of ‘professional control’ where, as noted earlier, they made assumptions about which stroke survivors were suitable for SSM and which were not. As a result, SSM may have been perceived as being less customizable to stroke survivors with greater complexity or impairments.

“It depends on the patient because we are primarily over 65′s…a lot of them are widows or on their own and some don’t have families. I kind of think (I will be) softer and just do it for them sort of thing rather than encouraging them to (self-management) you know? We would probably just do it for them you know?”(N1)

“To try and keep a stroke patient’s concentration for long is very difficult because I find myself, after about thirty minutes, you have lost them because they can’t concentrate for any long period of time. And if you’ve got patients with communication problems, which there is quite a lot of, I think they’d be put off. Sometimes people are very unrealistic; the stroke has maybe damaged something that has made them think that they can do something, and they can’t. If it’s safe, then you will let them try and if it’s not safe then we need to come up with the information so they start understanding why they can’t do something.”(N5)

## 4. Discussion

There is growing evidence of the effectiveness of SSM in helping stroke survivors and families to address the longer-term impact and ‘treatment burden’ associated with stroke [6,7,8,9,21,22]. Across Europe, there are calls for greater provision for, and further research on, frameworks to support the consistent implementation of longer-term care and integration of SSM into stroke pathways and service provision [14]. Understanding the challenges associated with embedding SSM into routine stroke service provision, as reported on in this paper, and how the different parts of the ‘system’ interact to facilitate, impede, or adapt to its implementation, is therefore important and timely. The findings from this secondary analysis reveal that the provision and implementation of stroke SSM by nurses remains professionally controlled to a large extent. Previous research highlights that this is by no means unique to nurses working in stroke rehabilitation settings [7,20,33,34,35,36], which offers confidence in the transferability of the findings of this secondary analysis.

The predominance of professional control over SSM and the challenges that practitioners, such as nurses, experience in implementing SSM in practice can be explained by a number of factors operating at multiple levels of the healthcare system. SSM encourages practitioners to work differently and more collaboratively with people with long-term conditions and in ways that focus on a person-driven agenda. However, this conceptualization jars with how the system, and health professionals, currently operate within a biomedical model. Indeed, as reported here, the nurses in this study still perceived SSM as being rooted in traditional notions of ‘compliance’ or ‘adherence’ with professionally determined goals rather than an approach which is based on working in partnership with stroke survivors to jointly agree on and address the SSM priorities and needs that are important to individuals. Such perspectives constrain the provision of SSM, which aligns with individuals’ priorities and capacity and can erode practitioners’ confidence in reflecting and challenging the ‘status quo’ of their practice and provide SSM beyond a medical paradigm [37,38]. These findings mirror and contribute to a growing literature base on professionals’ perceptions and conceptualizations of person-centred care in addressing the complexity of long-term condition management [20,33,34,35,36,37,38,39].

Self-management was also viewed by nurses as a ‘risky’ behavior, or only suitable for stroke survivors who were sufficiently motivated or had less severe impairments. This shows some evidence of what other studies have referred to as the prioritization of expert-based knowledge and health professionals’ tendency to dichotomize individuals into ‘good’ or ‘bad’ self-managers [18,37,39]. These views could differ, however, with how stroke survivors and families choose to engage in self-management (e.g., trial and error, risk taking) [36], making implementation a challenge. The findings revealed a sense of resistance towards SSM and a disconnect between nurses’ sense of professional purpose (i.e., to fix and to do for) and their perceived roles in ‘facilitating’ SSM, which has been found in other research on practitioners’ roles and perspectives in the field of self-management and health promotion [39]. The way in which community nursing follow-up was delivered within the organization meant that SSM was frequently positioned as an alternative to formal nursing care or synonymous with ‘going it alone’ and being discharged. Thus, the organization of longer-term nursing care in community settings may have constrained how the models of SSM are conceptualized, understood and implemented in practice. It may come as little surprise, therefore, that confusion exists amongst nurses on how they can best support people to self-manage within this model.

The complexity of the system, influenced by the external infrastructure and the organization itself, may limit the extent to which nurses have the confidence and capacity to deliver on a person-centred agenda of SSM in stroke care and does little to create the conditions that can enable nurses to practice their person-centred skills or to tailor care towards individualized need rather than the needs of the service. Our findings highlight the very real professional concerns, fears and tensions that need to be addressed and managed across multiple layers of external and internal context in the implementation of true person-centred SSM and to help move it away from it being so professionally driven. There is a need to create a positive culture of support for self-management to create a ‘safe’ environment for staff to feel confident and competent to learn about, practice and implement new models of SSM for stroke survivors and families, and where person-centred values are not in opposition to a dominant biomedical, organizational perspective. There is also a need to further conceptualize and clarify what SSM is and what it comprises in order to address some of the practitioners’ fears and tensions and to encourage them to see their roles in SSM as being congruent with their wider practice.

It is acknowledged that the data used in this secondary analysis represent the views of a small sample of nurses who participated in the original study that was designed to evaluate a specific SSM intervention for stroke survivors rather than the implementation of SSM in practice in general [11]. It may be that nurses with different demographic characteristics to those included in this study or from different geographical and organizational contexts not included in the original study may generate different views and interpretations. Nonetheless, as the findings reported here are consistent with other professional groups [20,33,34,35,36].

The original study, from which the data were drawn, was completed in 2014. It is plausible that since this time, the developments in the political landscape surrounding, and the culture of support for, self-management more broadly, have increased. This, coupled with the evolution of digital health in stroke (and in other neurological conditions) such as smart phone applications, telerehabilitation, and telemonitoring [40,41,42,43] for supporting rehabilitation and long-term behavior change, may have impacted on and shifted perceptions about SSM and its implementation somewhat. A secondary analysis approach potentially limited the extent to which broader issues of implementation could have been probed for further exploration in the original focus groups/interviews meaning that some questions may remain unanswered. However, the focus of the original study and its research questions were close enough to the secondary analysis research questions to tentatively explore broader issues around implementing SSM in practice. Both the original study and the secondary analysis of the dataset were undertaken by the same research team, reducing the risk of decontextualization of the data in the secondary analysis [44].

We used Lau et al.’s [27] categorization to frame this secondary analysis and although it was a useful implementation science model, as identified in other research [45], we did find some confusing overlap between the levels of the framework and at times struggled to code these to one part of the framework only. There were some parts of the framework that we found no information about, for example, how incentives, technological advances, skill mix and division of labor, and competencies influenced the implementation of SSM in stroke practice, and less information within the intervention category, so some further research to explore these domains may be valuable.

The limitations notwithstanding, the key message from this secondary analysis is that further consideration needs to be given to addressing the very real challenges associated with implementing SSM into existing healthcare systems, characterized by complexity and professional dominance. If SSM continues to be implemented with little sensitivity to the factors that influence, shape or constrain how SSM is interpreted and implemented, it will be less effective and have little meaningful and sustained impact on addressing the longer-term needs, priorities and outcomes of stroke survivors. An understanding of SSM needs and the perspectives of stroke survivors and their families are, of course, critical to this, and the research reported on here provides a complementary understanding of the challenges faced in the delivery and implementation of person-centred SSM for stroke survivors. The findings reported here have transferability to other long-term conditions, different multidisciplinary practitioners, and other countries around the world. Future research addressing the contexts that are conducive to triggering the mechanisms and outcomes underpinning the implementation of SSM and addressing the challenges in creating the conditions, cultures and environments that enable practitioners to consistently offer person-centred SSM in a time- and resource-stretched health and social care system would be valuable. Attempts to narrow the gap between the policy on, and implementation of, SSM will help to ensure that stroke survivors and families benefit from SSM in the longer-term.

## Figures and Tables

**Figure 1 jcm-09-00985-f001:**
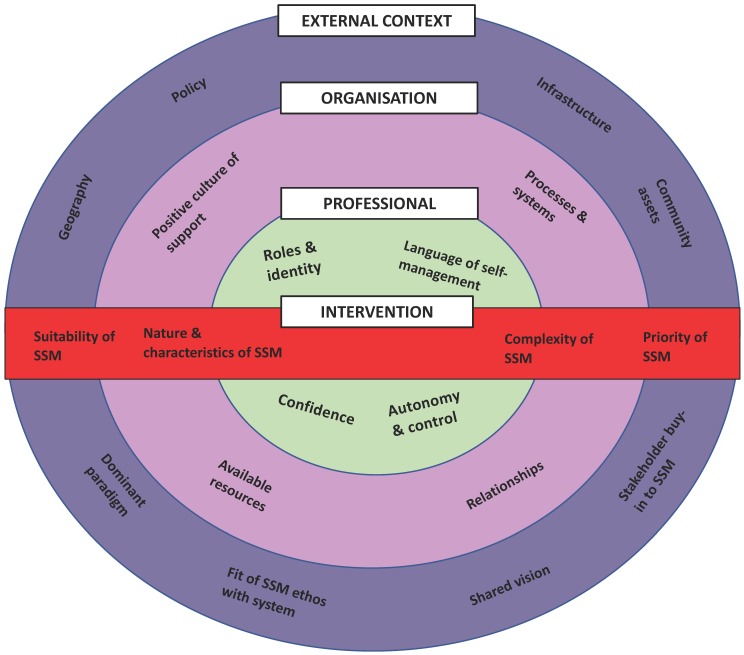
Conceptual framework describing key elements that influence the implementation of change in primary care (Lau et al., 2016) [27].

**Table 1 jcm-09-00985-t001:** Participant demographics.

Participant ID	Sex	Length of Time Qualified	Number of Years in Current Post	Highest Educational Qualification
N1	Female	11 years	3 years	Bachelor’s Degree
N2	Female	25 years	8 years	Diploma in Nursing
N3	Male	38 years	13 years	Bachelor’s Degree
N4	Female	17 years	2 years	Master’s Degree
N5	Female	16 years	9 years	Bachelor’s Degree
N6	Female	12 years	4 years	Bachelor’s Degree
N7	Female	21 years	9 years	Bachelor’s Degree
N8	Female	28 years	<1 year	Diploma in Nursing
N9	Female	22 years	6 years	Bachelor’s Degree
N10	Female	26 years	10 years	Diploma in Nursing
N11	Male	25 years	8 years	Master’s Degree
N12	Female	11 years	5 years	Bachelor’s Degree

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
