# Peer review of "Implementing Supported Self-Management in Community-Based Stroke Care: A Secondary Analysis of Nurses’ Perspectives"

_jcm, 2020, doi:10.3390/jcm9040985_

Round 1
Reviewer 1 Report
The issue of supported self management in community based stroke care is an important one for policy and practice. This study involved secondary analysis of qualitative data from 16 community stroke nurses collected in 3 focus groups and 2 interviews in 2014. There area a number of issues which need to be addressed in the paper:
- There have been changes since 2014 that should be acknowledged in the discussion especially the introduction of e-health into community-based rehabilitation of stroke.
- There is a brief reference to community rehabilitation but there should be more discussion about the interface between rehabilitation and supported self management including the uptake of rehabilitation prior to supported self management.
- The reference to the previous study was redacted to blind this reviewer. Thus, it is not possible to determine how this secondary analysis differed from the original analysis and study.
- There is insufficient information about the process of the focus groups and interviews – including how conducted (who chaired etc), how recorded, duration of focus groups etc.
- There is also insufficient information about the measures taken in the analysis to achieve rigor and if thematic saturation was reached.
- The results begin with Table 1. This is very long (12 pages) and difficult to read. The themes and subthemes are too brief and the examples have to be read to get an idea of what these mean. The quotes would be better illustrating not the themes but rather the analysis in section 3.1. For example the authors note: “Despite the intensifying political support for SSM at the time of the study, there was a sense amongst many of the nurses that SSM wasn’t anything new to them”. The quotes in the table illustrate this. The authors go on to give shorter excerpts of quotes mixed in with the text in 3.1. I found myself having to flip back and forth comparing what was said in the text and table 1.
- In the results under organisation the discussion would benefit from a clearer distinction between rehabilitation and SSM especially in the discussion around discharge. It would also be useful to more clearly distinguish SSM from other aspects of nursing care for stroke survivors in the community.
- This study represents the viewpoint of nurses. It is important to acknowledge the importance of the perspectives of stroke survivors, their families and carers.
Author Response
Reviewer 1
Point 1: There have been changes since 2014 that should be acknowledged in the discussion especially the introduction of e-health into community-based rehabilitation of stroke.
Response: We have now specifically highlighted the introduction of digital health in community-based stroke rehabilitation (Page 10, lines 429-434). We have also added new supporting references for this (references 40-43).
Point 2: There is a brief reference to community rehabilitation but there should be more discussion about the interface between rehabilitation and supported self-management including the uptake of rehabilitation prior to supported self-management.
Response: Much confusion exists still as to where SSM ‘fits’ in rehabilitation pathways and the overlap, or distinctions between, rehabilitation and SSM as the reviewer’s comment alludes to. We hope that we have interpreted the reviewer’s comments correctly and, in our response, addressed the challenges of this interface further in our revisions (Page 2, lines 51-72). We have added further references to support the revised text (references 23 and 24).
Point 3: The reference to the previous study was redacted to blind this reviewer. Thus, is it not possible to determine how this secondary analysis different from the original analysis and study.
Response: We have provided further clarity the nature of the original study on Page 3, lines 95-121 (the text has also changed in response to points 4 and 5 from Reviewer 1 and points 3 and 4 from Reviewer 2). For the purposes of transparency, the original study reference is Kidd, L.; Lawrence, M.; Booth, J.; Rowat, A.; Russell, S. Development and evaluation of a nurse-led, tailored stroke self-management intervention. BMC Health Serv Res 2015, 15, 359 doi: 10.1186/s12913-015-1021-y.
Point 4: There is insufficient information about the process of the focus groups and interviews – including how conducted (who chaired etc), how recorded, duration of focus groups etc
Point 5: There is also insufficient information about the measures taking in the analysis to achieve rigor and if thematic saturation was achieved.
Response to Point 4 and 5: We have now included additional information regarding the conduct and recording of the focus groups and measures taken to address the issue of rigour and saturation (Pages 3 & 4, lines 109-129).
Point 6: The results begin with Table 1. This is very long (12 pages) and difficult to read. The themes and subthemes are too brief and the examples have to be read to get an idea of what these mean. The quotes would be better illustrating not the themes but rather the analysis in section 3.1. For example the authors note: “Despite the intensifying political support for SSM at the time of the study, there was a sense amongst many of the nurses that SSM wasn’t anything new to them”. The quotes in the table illustrate this. The authors go on to give shorter excerpts of quotes mixed in with the text in 3.1. I found myself having to flip back and forth comparing what was said in the text and table 1.
Response: We have now removed the original Table 1 and included the more detailed example quotations in the text within sections 3.1, 3.2, 3.3 and 3.4 (Pages 4-9, lines 149-364). We hope that this addresses the reviewer’s comment and has strengthened the reporting in the results section. We are happy to revise this if we have mis-interpreted the reviewer’s comment on this point.
Point 7: In the results under organisation the discussion would benefit from a clearer distinction between rehabilitation and SSM especially in the discussion around discharge. It would also be useful to more clearly distinguish SSM from other aspects of nursing care for stroke survivors in the community.
Response: It was difficult to identify from the analysis how nurses distinguished between SSM and other aspects of their nursing care and indeed, we did not specifically look for this during the secondary analysis. We acknowledge it is an important point and indeed reflects one of the main purposes for undertaking this secondary analysis i.e. that there is significant variation in how nurses interpret SSM and its place in community-based stroke service delivery. We have attempted to clarify this in the text (Page 5, lines 156-163). We have also attempted to provide a clearer distinction between the phases of rehabilitation and SSM in relation to discharge (Page 6, lines 237-248). Again, we are happy to re-examine this if we have mis-interpreted the reviewer’s comments on this point.
Point 8: This study represents the viewpoint of nurses. It is important to acknowledge the importance of the perspectives of stroke survivors, their families and carers.
Response: We wholeheartedly agree with this comment and acknowledge the importance of understanding stroke survivors, families and carers perspectives of self-management. We see our paper as complementing existing research which has attempted to explore SSM from the perspective of stroke survivors and their families so as to offer some understanding of the additional challenges from a professional and ‘system’ side that influence the delivery of SSM. We have added text to clarify this in the text (Page 10, lines 456-459). We can confidently reassure the reviewer that understanding and supporting self-management from stroke survivors’ perspectives remains a core feature of our research.
Reviewer 2 Report
The authors have presented a qualitative study of the secondary analysis of nurses' perspectives of implementing supported self-management in community-based stroke care. The paper is well written and a joy to read. However, there are a few comments listed below that may be considered by the authors to further improve the quality of the paper.
Title: I recommend that 'stroke' before nurses' be deleted in line 4.
Introduction: Line 34, spell out 'UK' before the first use of abbreviation. Line 61, delete 'in' after implementation. Lines 69, change the word 'aim' to objective, and in line 71, change 'intention' to aim. Aims are measurable aspects of the study objective.
Methods: It will be easier to read this section if it has sub-headings. I recommend that the narrative be placed under sub-headings that include Study participants, Data source, Study instruments (focus groups and telephone interviews), Data analysis.
It was stated that data were collected via the focus groups and interviews and was completed in 2014. State when data collection started. Include in the methods section how many focus groups and telephone interviews were conducted respectively. Also give examples of questions that were asked during the interviews.
How was qualitative analysis done? Was any statistical software like NVivo utilized?
Results: It is better to organize this section with accompanying tables into participants' demographics/characteristics, and other results from the data analysis. Table 1 should depict the age, years of experience, sex and highest qualification of the nurses. These characteristics may influence the perspectives of the study participants.
The current Table 1 should be changed to Table 2, and it is preferable to include the grid lines in order to give the table a nicer appearance. It is also better to add the table after the narrative of the contents of the conceptual framework were explained; i.e., after lines 115-199.
Discussion: Lines 227-228, correct 'what others' to what other, and revise the sentence to "... dichotomize individuals into ‘good’ or ‘bad’ self-managers".
Minor comment: Check for other 'typos' and minor errors in grammar.
Best of luck!
Author Response
Reviewer 2
Point 1: Title: I recommend that 'stroke' before nurses' be deleted in line 4.
Response: This has now been removed to read ‘Implementing supported self-management in community-based stroke care: a secondary analysis of nurses’ perspectives (Page 1, Line 4)
Point 2: Introduction: Line 34, spell out 'UK' before the first use of abbreviation. Line 61, delete 'in' after implementation. Lines 69, change the word 'aim' to objective, and in line 71, change 'intention' to aim. Aims are measurable aspects of the study objective.
Response: Line 34 has been changed to spell out United Kingdom in full. Requested word changes in lines 61, 69 and 71 have now been made (Page 2, now lines 82 and 84).
Point 3: Methods: it will be easier to read this section if it has sub-headings. I recommend that the narrative be placed under sub-headings that include Study participants, Data source, Study instruments (focus groups and telephone interviews), Data analysis.
Response: We have now used the following subheadings to split up the narrative: study design and data source, study participants, study instruments (focus groups and telephone interviews), data analysis, and secondary analysis.
Point 4: It was stated that data were collected via the focus groups and interviews and was completed in 2014. State when data collection started. Include in the methods section how many focus groups and telephone interviews were conducted respectively. Also give examples of questions that were asked during the interviews.
Point 5: How was qualitative analysis done? Was any statistical software like NVivo utilized?
Response to Point 4 and 5: We have now stated when data collection started (Page 3, line 110). We have also added how many focus groups and telephone interviews were conducted (Page 3, lines 110-111) and provided some examples of the questions that were asked (Page 3, lines 117-121). The qualitative analysis was done manually without using NVivo based on personal preference. This has been clarified (Page 3, line 123).
Point 6: Results: It is better to organize this section with accompanying tables into participants’ demographics/characteristics, and other results from the data analysis. Table 1 should depict the age, years of experience, sex and highest qualification of the nurses. These characteristics may influence the perspectives of the study participants.
Response: Whilst we didn’t collect demographic information from participants about their age, we are able to present information about the sex, number of years qualified, number of years in current post and highest educational qualification of the nurses who participated. We have now presented these in Table 1. We agree with the reviewer’s point that the characteristics of our sample may influence the participants’ perspectives and have more clearly emphasised this in the discussion section (Page 10, lines 426-427). We have also amended the number of participants from 16 to 14 as when we went back to the demographic data (in response to point 6 from Reviewer 2), we realised that we had counted two individuals had been counted twice incorrectly in the sample size (since they took part in both the pre and post-intervention phase).
Point 7: The current Table 1 should be changed to Table 2, and it is preferable to include the grid lines in order to give the table a nicer appearance. It is also better to add the table after the narrative of the contents of the conceptual framework were explained; i.e., after lines 115-199.
Response: In response to point 6 from Reviewer 1, we have now removed the table and subsumed the example quotations within the narrative.
Point 8: Discussion: Lines 227-228, correct 'what others' to what other, and revise the sentence to "... dichotomize individuals into ‘good’ or ‘bad’ self-managers".
Response: These corrections have now been made (page 9, line 394-397).
Point 9: minor comment: check for other typos and minor errors in grammar
Response: we have proofread the paper for minor typos and errors in grammar and have corrected these.
Round 2
Reviewer 1 Report
The authors have adequately addressed the comments in my previous review. No further comments.
Reviewer 2 Report
The authors have satisfactorily addressed my concerns and comments from the first review.